# S4D: Streaming 4D Real-World Reconstruction with Gaussians and 3D Control Points

## Abstract

Dynamic scene reconstruction using Gaussians has recently attracted increased interest. Mainstream approaches typically employ a global deformation field to warp a 3D scene in canonical space. However, the inherent low-frequency nature of implicit neural fields often leads to ineffective representations of complex motions. Moreover, their structural rigidity can hinder adaptation to scenes with varying resolutions and durations. To address these challenges, we introduce a novel approach for streaming 4D real-world reconstruction utilizing discrete 3D control points. This method physically models local rays and establishes a motion-decoupling coordinate system. By effectively merging traditional graphics with learnable pipelines, it provides a robust and efficient local 6-degrees-of-freedom (6-DoF) motion representation. Additionally, we have developed a generalized framework that integrates our control points with Gaussians. Starting from an initial 3D reconstruction, our workflow decomposes the streaming 4D reconstruction into four independent submodules: 3D segmentation, 3D control point generation, object-wise motion manipulation, and residual compensation. Experimental results demonstrate that our method outperforms existing state-of-the-art 4D Gaussian splatting techniques on both the Neu3DV and CMU-Panoptic datasets. Notably, the optimization of our 3D control points is achievable in 100 iterations and within just 2 seconds per frame on a single NVIDIA 4070 GPU.

## 1 Introduction

Reconstructing real-world scenes is a longstanding challenge in computer graphics. Recently, 3D Gaussian Splatting (3D-GS) Kerbl et al. (2023) have demonstrated remarkable success in producing high-quality reconstructions for static scenes. This technique utilizes discrete Gaussians to model the scene, attributing them with physically meaningful properties, and achieves rendering by "splatting" these Gaussians onto the image plane. For dynamic scene reconstruction, current Gaussian Splatting methods Yang et al. (2023b); Huang et al. (2023); Wu et al. (2023); Lin et al. (2023b) build upon the concept introduced by dynamic NeRFs Park et al. (2021a;b); Pumarola et al. (2021). These methods typically decompose scene motion into a canonical space and employ an implicit neural field to capture global motion. However, the low-frequency characteristics and rigid structure of implicit neural networks limit their ability to accurately handle complex motions or scenes with diverse resolutions and varying lengths.

Compared to NeRF, one significant advantage of 3D-GS lies in its discrete structure. This structure ensures that scene representation—via Gaussians—is concentrated at influential positions within the scene, avoiding the inefficiencies of a global field that waste representational capacity on empty space. Similarly, the global implicit neural field used for deformation faces a similar issue: only a small part of the scene is dynamic, while the majority remains static. Therefore, a discrete, localized motion representation holds promise, as it offers precise and flexible modeling of 3D motions at a local level.

Previous methods, both traditional Vedula et al. (1999) and learning-based Huang et al. (2023), have attempted to represent local 3D motion in a discrete fashion. However, traditional methods often struggle with complex 3D alignment challenges, while learning-based approaches face difficulties in convergence due to their high degrees of freedom (DoF).

Our approach differs by integrating graphics and learnable pipelines. Recognizing that optical flow is the 2D projection of scene flow, and that optical flow estimation Ranjan & Black (2017); Sun et al.

(2018); Teed & Deng (2020); Kroeger et al. (2016) is a well-established technique, we simplify the problem by introducing a novel local decoupling system. In this system, local 3D motion is decoupled into the observable and hidden components. The observable component is tied to optical flow, while the hidden component remains learnable. We refer to this new form of motion representation as the "3D control point" approach.

Building on the introduced 3D motion representation, we present a novel generalized streaming framework for 4D real-world reconstruction. Beginning with an initial 3D reconstruction, our workflow breaks the streaming 4D reconstruction process into distinct submodules: 3D segmentation, 3D control point generation, object-wise motion manipulation, and residual compensation. This structured approach helps to minimize topological errors and ensures a compact, robust representation.

Our key contributions are as follows:

- We model the 3D motions of dynamic objects discretely using an innovative method that combines graphics techniques with learnable pipelines. This approach decouples the components of 3D motion: observable part is tied to optical flow, while the hidden part is learned. This motion representation, referred to as "3D control points," improves both convergence speed and reconstruction accuracy.
- We introduce a generalized streaming pipeline that employs Gaussians and 3D control points to reconstruct 4D real-world scenes. Our method establishes a new benchmark, achieving state-of-the-art performance on the Neu3DV and CMU-Panoptic datasets.

## 2 RELATED WORK

### 2.1 4D RECONSTRUCTION

The recent Neural Radiance Field (NeRF) Mildenhall et al. (2021) approach has proven efficient for scene reconstruction by utilizing a global continuous implicit representation of the scene. Subsequent works have enhanced reconstruction quality by evolving from MLPs to grid-based structures. For instance, Müller et al. (2022) introduced the Instant Neural Graphics Primitives method, while Fridovich-Keil et al. (2022) proposed Plenoxels. Other methods, like Mip-NeRF Barron et al. (2021; 2022), modeled rays as cones to achieve anti-aliasing.

To accelerate rendering, various strategies have been proposed, such as pre-computation Wang et al. (2023c; 2022); Fridovich-Keil et al. (2022); Yu et al. (2021) and hash coding Müller et al. (2022); Takikawa et al. (2022).

Efforts to extend NeRF's representation to dynamic scenes have also been explored. For instance, Xian et al. (2021); Wang et al. (2021) challenged the static scene hypothesis by providing separate representations over time. Mainstream methods Liu et al. (2022); Park et al. (2021a;b); Pumarola et al. (2021); Song et al. (2023); Du et al. (2021); Li et al. (2021); Liu et al. (2023) decoupled motion from the scene using a deformation field to warp the 3D scene at a canonical timestep.

Further advancements included expanding the grid Fang et al. (2022); Shao et al. (2023) and planar Cao & Johnson (2023); Fridovich-Keil et al. (2023) structures with a time dimension to boost rendering speed and reconstruction quality. Other research Wang et al. (2023b); Li et al. (2023); Lin et al. (2022; 2023a) focused on refining reconstruction quality by leveraging prior knowledge of camera positions. Additional supervision information such as depth Attal et al. (2021) and optical flow Wang et al. (2023a) was incorporated during training. NeRFPlayer Song et al. (2023), notably, utilized self-supervised learning to segment dynamic scenes into static, deforming, and newly appearing regions, applying tailored strategies to each.

Recently, 3D-GS Kerbl et al. (2023) introduced an elegant point-based rendering approach with efficient CUDA implementations. The 4D reconstruction methods using Gaussians closely resemble those in NeRF. For instance, Yang et al. (2023a) incorporated time-variant attributes into Gaussians, while Dynamic-GS Luiten et al. (2023) directly learned dense Gaussian movements. 3DGStream introduced Neural Transformation Cache to represent Gaussian motion on a frame-by-frame basis, and Gaussian-Flow Lin et al. (2024) employed Dual-Domain Deformation Model to represent point-wise movement. HiFi4G Jiang et al. (2023) combined an implicit method Wang et al. (2023d) for surface reconstruction with a traditional graphics approach Sumner et al. (2007) for surface warping.

Other studies Wu et al. (2023); Yang et al. (2023b); Huang et al. (2023); Lin et al. (2023b) represented 4D scenes by warping Gaussians in a canonical space using global implicit deformation fields.

## 2.2 3D MOTION REPRESENTATION

Decoupling 3D motion from the canonical scene offers significant advantages by reducing redundant information in the time dimension. While implicit neural representations are compact, they often lack the flexibility needed to adapt to varying scenes and require redesigns. Traditional graphics methods Sorkine (2005); Yu et al. (2004) have provided flexible deformation solutions while preserving geometric details. Among these, Sumner et al. (2007) introduced sparse control points (ED-graph) to represent the motion of dense surfaces, balancing compactness and flexibility.

HiFi4G Jiang et al. (2023) utilized the ED-graph approach directly, but its surface reconstruction demands dense camera distribution and is time-consuming. SC-GS Huang et al. (2023) also adopted the concept of sparse control points, yet direct optimization is difficult due to the high degrees of freedom. Moreover, unlike meshes, the volumetric radiance representation of 3D Gaussians is not well-suited for the thin nature of surfaces Huang et al. (2024).

To address these challenges, we propose a novel 3D control points method tailored for volumetric radiance representations. By decoupling 3D motion into observable and hidden components, we apply distinct strategies for motion acquisition. This approach leverages the strengths of traditional graphics techniques and learnable pipelines, enabling precise and flexible motion representation while maintaining the efficiency and compactness necessary for further applications.

## 3 METHODOLOGY

The streaming workflow is illustrated in Fig. 1. Our method consists of four independent modules: 3D segmentation, 3D control point generation, object-wise motion manipulation, and residual compensation. Inputs include 3D Gaussians, 2D masks, and backward optical flow. The 3D Gaussians $\mathbb{G}_{t-1}$ are obtained either from the initial static scene reconstruction using 3D-GS Kerbl et al. (2023) or from the previous frame. The 2D object masks, $\mathbf{M}_{t-1}$, are generated via the SAM-track method Cheng et al. (2023). Optical flows, $\mathbf{F}_{t-1}$, are derived from established optical flow methods Kroeger et al. (2016); Ranjan & Black (2017); Sun et al. (2018). Section 3.1 provides a brief overview of each module, with detailed explanations following in subsequent sections.

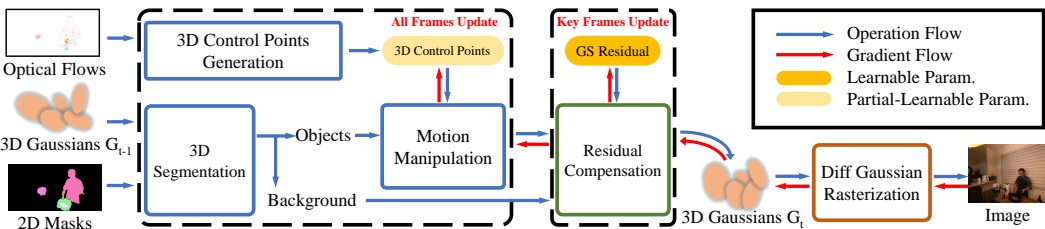

Figure 1: **Streaming workflow.** The workflow starts by segmenting the scene into a static background and several moving objects using multiview masks combined with a Gaussian category voting algorithm. Optical flow is then applied to create a partially learnable system of 3D control points. Motion-related attributes of the Gaussians are manipulated on an object-wise basis. To prevent reconstruction failures from error accumulation, Gaussian attributes are periodically updated in a keyframe manner, capturing additional scene information as attribute residuals of the Gaussians.

## 3.1 OVERVIEW OF EACH MODULE

**3D Segmentation.** The goal of 3D segmentation is to label each Gaussian as either a moving object or part of the static background. As outlined in Sec. 1, representing local motion discretely by an object-wise approach is often more efficient than global methods. Consequently, defining the regions where each local representation is applicable is crucial. To achieve this, we use multiview masks and adopt a Gaussian category voting algorithm to divide the scene into dynamic objects and a static background. Only Gaussians assigned to specific objects are influenced by the 3D control points.

**3D Control Points Generation.** The goal of 3D Control Points Generation is to generate 3D control points for all moving objects, which govern the motion of their associated Gaussians on an object-wise basis. This module plays a pivotal role in our workflow. Direct optimization of 3D control points is challenging due to their high degrees of freedom. To address this issue, we introduce a local decoupling coordinate system that links partial parameters of the 3D control points with 2D optical flow. This innovative approach effectively reduces the control points' degrees of freedom from nine to three, enabling faster convergence during optimization.

**Object-wise Motion Manipulation.** The goal of Object-wise Motion Manipulation is to control the movement of Gaussians for each object using the 3D control points. Motion manipulation involves transforming the position and rotation attributes of Gaussians from one timestep $t-1$ to the next $t$. By manipulating these attributes on an object-wise basis, each Gaussian is influenced only by the control points associated with its category. This selective control method allows for accurate handling of topology changes of spatially adjacent objects and improves the precision of discrete motion representation.

**Residual Compensation.** The goal of Residual Compensation is to mitigate error accumulation and ensure stable long-term reconstruction. This module activates only at keyframes. Specifically, during non-keyframe intervals, Gaussian attributes are frozen, and updates are confined to the control points. At keyframes, both Gaussian attributes and control points are optimized, enabling comprehensive adjustments that preserve the continuity and accuracy of the scene reconstruction. It is important to note that the term "residual" in this context refers to adjustments in the attributes of the existing Gaussians, without the introduction of new Gaussians.

## 3.2 3D SEGMENTATION

A straightforward and efficient method for 3D segmentation is to leverage objects' 2D masks combined with a statistical method to categorize Gaussians based on their projections from multiple viewpoints. We utilize the SAM-track method Cheng et al. (2023) to obtain continuous 2D object masks and implement a Gaussian category voting strategy similar to the approach used in SA-GS Hu et al. (2024). A notable improvement in our method is its ability to distinguish between multiple objects in the foreground.

Specifically, each Gaussian is assigned a counter for every object category, with the total number of counters corresponding to the number of 2D object masks. For each Gaussian $\mathbf{g}$, we project it onto the $j^{th}$ image plane. If the projection of the Gaussian falls within the boundary of the $k^{th}$ object's mask, the counter for that object's category is incremented by one. This process is repeated across all training views, denoted as $C_k$. After processing all views, we label each Gaussian as $\mathbf{L}$ based on the category with the highest counter value, indicating its assigned object category. The complete algorithm is summarized as follows:

$$
\begin{aligned}
C_k &= C_k + 1 \quad \text{if } \mathbf{P}_j \mathbf{g} \in \mathbf{m}_{jk}, \\
\mathbf{L} &= \text{argmax}(\mathbf{C}).
\end{aligned}
\tag{1}
$$

## 3.3 3D CONTROL POINTS GENERATION AND LOCAL 6-DOF MOTION DECOUPLING

To effectively represent local 6-degrees-of-freedom (6-DoF) motion, the attributes of a 3D control point must include its position, local spatial translation, and rotation. However, only a portion of this motion is visible from a single viewpoint, which is captured as optical flow. To address this limitation, we develop an advanced coordinate system termed the "ray coordinate system." This system correlates observable 3D motion components with optical flow from a single viewpoint. Motion components hidden from this viewpoint are optimized using global image loss, as their information is implicitly captured across multiple views.

To streamline the representation of motion, we model a small cluster of localized rays from the scene to the camera as parallel lights. This approach enables the rays to share a unified coordinate system, focusing on local motion rather than individual point-wise motion. The reasoning behind approximating these rays as nearly parallel is explained in Appendix .1. In this model, the z-axis of the ray coordinate system aligns with the center ray of the cluster, which extends from the camera towards the scene. The motion decoupling is depicted in Fig. 2(a) and Fig. 2(b).

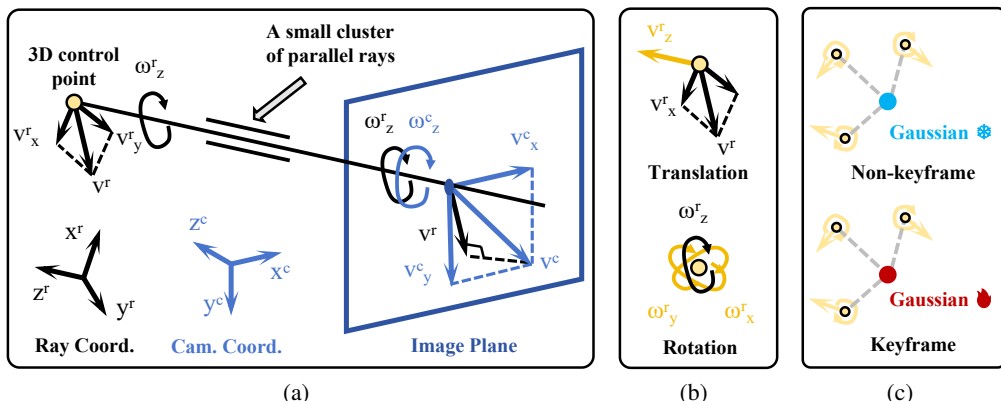

(a)    (b)    (c)

Figure 2: **3D Control Points.** (a) We model the localized light as parallel rays. The motion parameters of the 3D control points, represented by the black segments, are bound with local 2D motion priors derived from the optical flow on the image plane. (b) The yellow segments remain learnable because a small cluster of parallel rays cannot adequately represent these elements. To decouple these attributes, we use Euler angles to represent rotational components. (c) The 3D control points spatially influence adjacent Gaussian points. During non-keyframe intervals, the Gaussian attributes remain fixed, while only the learnable components of the control points undergo optimization. At keyframes, the attributes of the Gaussians are actively included in the optimization process. For effective interpolation, rotation attributes are expressed in quaternion format.

For translations, the component perpendicular to the ray affects the 2D optical flows observed in the image, while the component along the ray is not visible. Regarding rotation, when a cluster of rays is parallel, one rotational component aligns with the normal vector of the rays, practically translating perpendicular to them. This observable rotation appears in the 2D optical flows as rotation around the cluster's center. Other rotational components, corresponding to translations along the rays, are not detectable in the image plane.

A projection relationship is established between the 2D motion information on the image plane and the motion information carried by parallel light rays. Under the near-parallel hypothesis, points $\mathbf{x}_i$ within the neighborhood $\mathcal{N}$ of $\mathbf{x}_0$ share the ray coordinate system of $\mathbf{r}_0$, as depicted in Fig. 3(a). Fig. 3(b) and Eq. 2 detail the translation information $\mathbf{v}_r$ carried by rays near the image plane:

$$\mathbf{v}^r = f * \frac{1}{N} \sum_{\mathbf{x}_i \in \mathcal{N}} (\frac{\mathrm{d}x_i^r}{\mathrm{d}t}, \frac{\mathrm{d}y_i^r}{\mathrm{d}t})^{\mathrm{T}} = f * \frac{1}{N} \left[\mathbf{R}_{cr}\right]_{:2,:2} \left[\mathbf{K}^{-1}\right]_{:2,:2} \sum_{\mathbf{x}_i \in \mathcal{N}} (u_i, v_i)^{\mathrm{T}}, \quad (2)$$

where $(u_i, v_i)$ represents the optical flow of the points $\mathbf{x}_i$. The inverse intrinsic matrix $\mathbf{K}^{-1}$ converts the optical flow from pixel units to base units (meters), and the rotation matrix $\mathbf{R}_{cr}$ transforms the coordinate system from the camera to the ray coordinate system. The notation $[\cdot]_{:2,:2}$ refers to the matrix's first and second rows and columns. $N$ represents the number of pixels in the neighborhood $\mathcal{N}$. The camera's focal length, $f$, measured in meters, is used to convert the translation from the normalized to the physical image plane, enhancing the interpretability of $\mathbf{v}_r$. It is crucial to note that all measurements of $u, v$ are in pixels, while $X, Y, x, y, z$ are measured in meters.

Next, we back-project the translation information into the scene to obtain the control points' 3D positions and corresponding 2D translations, utilizing rough depth estimates derived from Gaussian rasterization. Specifically, the critical ratio for this process is the depth $z^c$ relative to the focal length $f$. The formulas for converting the translation and determining the position are presented in Equations 3 and 4, respectively:

$$\mathbf{V}^r = (\frac{\mathrm{d}X^r}{\mathrm{d}t}, \frac{\mathrm{d}Y^r}{\mathrm{d}t})^{\mathrm{T}} = \frac{z^c}{f} \mathbf{v}^r, \quad (3)$$

$$\mathbf{X}^c = \frac{z^c}{f}(x_0^c, y_0^c, f)^{\mathrm{T}}. \quad (4)$$

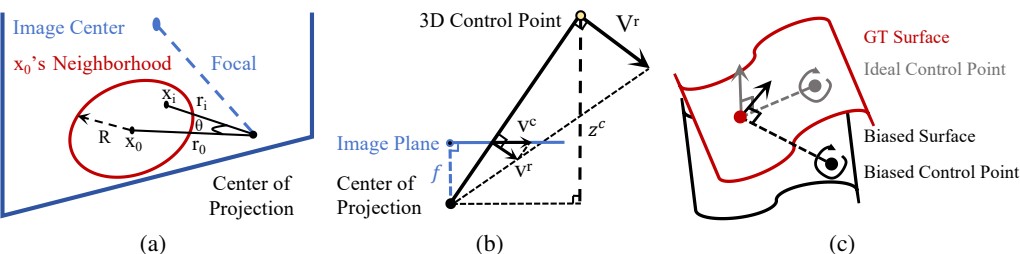

(a)  (b)  (c)

Figure 3: **Auxiliary Diagrams for Local Motion Mapping.** (a) An illustration of angles, points, and rays within the neighborhood of $\mathbf{x}_0$. (b) A quantitative depiction of motion projection. (c) A comparison between Gaussians distributed on the ground truth surface and control points located on a biased surface. The rough depth estimation can introduce biases in the predicted translations, particularly due to rotational movements.

To exploit the motion prior information provided by optical flow, we convert the rotation representation into the Euler angle format to facilitate decoupling. The formulation for angular velocity is specified as follows:

$$\boldsymbol{\omega}_z^r = \frac{1}{N} \sum_{\mathbf{x}_i \in \mathcal{N}} \frac{(\mathbf{x}_i^r - \mathbf{x}_0^r) \times \mathbf{v}_i^r}{\|\mathbf{x}_i^r - \mathbf{x}_0^r\|^2}. \tag{5}$$

Thus far, the motion decoupling process within the ray coordinate system is complete. We have obtained the 3D position and 2D translation of control points, along with 1D rotation information from a single viewpoint.

We also explored incorporating local macro rotation, as described in the ED-graph method Sumner et al. (2007). However, this resulted in a decline in performance. We hypothesize that the performance drop is due to structural differences between Gaussians and meshes: while meshes are tightly clustered on the object surface, Gaussians are more loosely distributed due to their higher degrees of freedom. Consequently, we omit the component concerning local macro rotation to minimize biases introduced by rough depth estimations, as illustrated in Fig. 3(c). Although this adjustment introduces some sparsity, it optimizes our motion manipulation approach for volumetric radiance representations like Gaussians.

### 3.4 OBJECT-WISE MOTION MANIPULATION

For each moving object, the translation $\mathbf{t}$ and rotation $\mathbf{q}$ of a Gaussian $G$ are determined by interpolating the translation $\mathbf{t}_i$ and rotation $\mathbf{q}_i$ of the K-nearest spatially adjacent control points $C_i$ within its neighborhood $\mathcal{N}$. The weights $w_i$ for this interpolation are inversely proportional to the Euclidean distance between the points. Rotations initially expressed in Euler angles are converted to quaternions to facilitate the interpolation process. For simplicity in the formulas, the normalization of quaternions to unit quaternions is omitted.

$$(\mathbf{t}, \mathbf{q}) = \sum_{C_i \in \mathcal{N}} w_i * (\mathbf{t}_i, \mathbf{q}_i), \tag{6}$$

$$w_i = \frac{1/\|\mathbf{X}_G - \mathbf{X}_{C_i}\|}{\sum_{C_i \in \mathcal{N}} 1/\|\mathbf{X}_G - \mathbf{X}_{C_i}\|}. \tag{7}$$

### 3.5 RESIDUAL COMPENSATION

Our approach utilizes control points to predict 3D motion but suffers from error accumulation over time due to two main factors. First, scene reconstruction relies on information from earlier frames, which may fail to capture previously occluded areas, such as the sides or backs of objects. Second, there are discrepancies between the predicted motion of the Gaussians and the ground truth, particularly for Gaussians distant from neighboring control points.

To mitigate error accumulation, we employ a strategy inspired by video coding techniques, specifically residual coding Wiegand et al. (2003); Sze et al. (2014). Residual coding in video technology compensates for differences between original and motion-compensated frames. Our model uses a keyframe updating method to optimize Gaussians after object-wise motion manipulation, thereby improving long-term streaming reconstruction.

As illustrated in Fig.2(c), during non-keyframes, Gaussian attributes remain fixed while only the learnable components of the 3D control points $\mathbb{C}$ are optimized. The scene at non-keyframe $t$ is expressed as follows:

$$\mathbb{G}_t = \mathcal{F}_{\mathbb{C}_t}(\mathbb{G}_{t-1}), \tag{8}$$

where $\mathcal{F}$ denotes object-wise motion manipulation. In keyframes, attributes of Gaussians are optimized, but instead of reinitializing them, they inherit attributes from previous timestamps. This approach enhances efficiency by updating only the attributes' residuals $\mathbb{R}$, which capture minor deviations. The scene at keyframe $t$ is represented as follows:

$$\mathbb{G}_t = \mathcal{F}_{\mathbb{C}_t}(\mathbb{G}_{t-1}) + \mathbb{R}_t, \tag{9}$$

We also introduce the concept of Group of Scenes (GoS), similar to the Group of Pictures (GoP) in video coding. Specifically, we segment the scene sequence into multiple groups of scenes (GoS), with each group comprising scene information across N consecutive timesteps. Given the initial 3D scene $\mathbb{G}_0$ with a GoS-N setting, the frame-by-frame representation for 4D scene evolves as follows:

$$\mathbb{G}_0, \mathbb{C}_1, \mathbb{C}_2, \ldots, \mathbb{C}_{N-1}, (\mathbb{R}_N, \mathbb{C}_N), \mathbb{C}_{N+1}, \mathbb{C}_{N+2}, \ldots \tag{10}$$

### 3.6 Loss Function

The loss function is straightforward, as shown in Eqn. 11:

$$\mathcal{L} = (1 - \lambda)\mathcal{L}_1 + \lambda\mathcal{L}_{\mathrm{D-SSIM}}, \tag{11}$$

where $\lambda$ is set to 0.2 as recommended by 3D-GS Kerbl et al. (2023). Our motion modeling and streaming pipeline is both efficient and robust, enabling our method to converge effectively without requiring additional loss constraints. This results from the inherent design of our system, which directly specifies the motion of the Gaussians from the structural framework, ensuring accurate and stable convergence.

## 4 Experiment

### 4.1 Datasets and Implementation Details

**Neu3DV Dataset Li et al. (2021).**   The Neural 3D Video Synthesis Dataset includes six sequences, originally captured at a resolution of 2704 × 2028, which were downsampled to 1352 × 1014 for training purposes. The sequence 'flame_salmon_1' contains 1200 frames, while the remaining five sequences consist of 300 frames each. All sequences were recorded using 15 to 20 static cameras, evenly distributed in a spherical configuration around the scene.

**CMU-Panoptic Dataset Joo et al. (2017).**   The CMU-Panoptic Dataset comprises three sequences that showcase complex, dynamic object motions. Each sequence has a resolution of 640 × 360 and consists of 150 frames. The data was collected using 31 static cameras, with 27 used for training and 4 reserved for testing, all placed to form a fanned-out arrangement in front of the scene.

**Implementation Details.**   All experiments were conducted on NVIDIA RTX 4070 GPUs. To assess our control point method for motion representation, we compared it with Dynamic-GS Luiten et al. (2023) and 4D-GS Wu et al. (2023), both maintaining a constant number of Gaussian points. Using their official codes, we ensured identical 3D point initialization across methods. Following 4D-GS guidelines, we used COLMAP Schönberger & Frahm (2016); Schönberger et al. (2016) to

initialize 3D points from the first training frames and adopted Dynamic-GS's 10k-iteration initial reconstruction. For subsequent frames, we trained with 500 iterations for keyframes and 100 for non-keyframes, using a single Adam optimizer with fixed learning rates as in Dynamic-GS. For the CMU-Panoptic Dataset, we reused the foreground mask provided by Dynamic-GS.

## 4.2 RECONSTRUCTION RESULTS

**Quantitative Results.**   We first present average quantitative results including PSNR, SSIM, and LPIPS in Table 1. Our method outperformed both Dynamic-GS Luiten et al. (2023) and 4D-GS Wu et al. (2023) across all metrics. Additionally, we provide per-scene quantitative results in Tables 2 and 3. our approach achieved state-of-the-art (SOTA) performance in terms of PSNR for most scenes and achieved superior SSIM and LPIPS scores for even more scenes, which reflects enhanced subjective reconstruction quality. 4D-GS's quantitative results on the CMU-Panoptic dataset were excluded because it struggled to handle violent scene motion, which led to object disappearance, as depicted in Fig. 4. Notably, our method demonstrated a significant advantage in training time.

Table 1: Average reconstruction results for the Neu3DV dataset. Training time is reported in hours. Each cell is color-coded to denote performance ranking: best for the top performance, second for the second best, and third for the third best.

| Metrics | PSNR↑ | SSIM↑ | LPIPS↓ | Training Time ↓ |
|---|---|---|---|---|
| Dynamic-GS | 27.65 | 0.9232 | 0.1313 | 57.35 |
| 4D-GS | 30.49 | 0.9401 | 0.0998 | 6.88 |
| Ours-GoS1 | 31.20 | 0.9468 | 0.0881 | 12.19 |
| Ours-GoS5 | 31.23 | 0.9459 | 0.0906 | 3.94 |
| Ours-GoS10 | 30.91 | 0.9437 | 0.0941 | 1.89 |

Table 2: Per-scene results for the Neu3DV dataset.

| Scene | *sear_steak* | | | *cook_spinach* | | | *cut_roasted_beef* | | |
|---|---|---|---|---|---|---|---|---|---|
| Metrics | PSNR↑ | SSIM↑ | LPIPS↓ | PSNR↑ | SSIM↑ | LPIPS↓ | PSNR↑ | SSIM↑ | LPIPS↓ |
| Dynamic-GS | 31.38 | 0.9469 | 0.1119 | 29.98 | 0.9388 | 0.1179 | 29.64 | 0.9360 | 0.1248 |
| 4D-GS | 31.62 | 0.9569 | 0.0808 | 32.79 | 0.9522 | 0.0926 | 32.13 | 0.9467 | 0.0959 |
| Ours-GoS1 | 33.23 | 0.9654 | 0.0719 | 33.20 | 0.9586 | 0.0796 | 33.00 | 0.9609 | 0.0795 |
| Ours-GoS5 | 33.72 | 0.9661 | 0.0704 | 32.91 | 0.9579 | 0.0819 | 33.23 | 0.9592 | 0.0835 |
| Ours-GoS10 | 33.64 | 0.9655 | 0.0716 | 32.65 | 0.9553 | 0.0861 | 32.47 | 0.9555 | 0.0890 |

| Scene | *flame_steak* | | | *flame_salmon_1* | | | *coffee_martini* | | |
|---|---|---|---|---|---|---|---|---|---|
| Metrics | PSNR↑ | SSIM↑ | LPIPS↓ | PSNR↑ | SSIM↑ | LPIPS↓ | PSNR↑ | SSIM↑ | LPIPS↓ |
| Dynamic-GS | 30.41 | 0.9429 | 0.1121 | 20.19 | 0.8875 | 0.1583 | 24.29 | 0.8870 | 0.1630 |
| 4D-GS | 29.28 | 0.9545 | 0.0836 | 28.27 | 0.9106 | 0.1289 | 28.87 | 0.9198 | 0.1168 |
| Ours-GoS1 | 32.84 | 0.9645 | 0.0723 | 28.00 | 0.9173 | 0.1083 | 26.90 | 0.9140 | 0.1170 |
| Ours-GoS5 | 33.18 | 0.9649 | 0.0707 | 27.65 | 0.9155 | 0.1127 | 26.71 | 0.9119 | 0.1242 |
| Ours-GoS10 | 32.94 | 0.9631 | 0.0733 | 27.17 | 0.9127 | 0.1165 | 26.51 | 0.9100 | 0.1283 |

Table 3: Per-scene results for the CMU-Panoptic dataset.

| Scene | *softball* | | | *boxes* | | | *basketball* | | |
|---|---|---|---|---|---|---|---|---|---|
| Metrics | PSNR↑ | SSIM↑ | LPIPS↓ | PSNR↑ | SSIM↑ | LPIPS↓ | PSNR↑ | SSIM↑ | LPIPS↓ |
| Dynamic-GS | 26.93 | 0.9076 | 0.1804 | 27.79 | 0.9069 | 0.1769 | 28.54 | 0.9032 | 0.1812 |
| Ours-GoS2 | 27.48 | 0.9264 | 0.1374 | 27.88 | 0.9227 | 0.1413 | 27.72 | 0.9203 | 0.1423 |

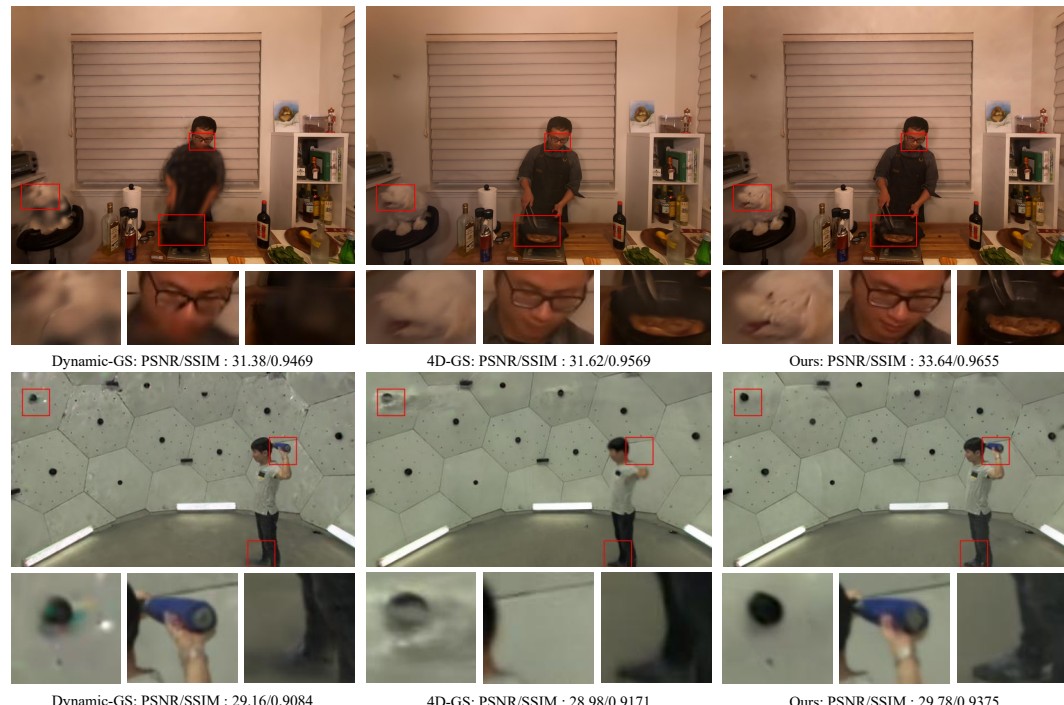

Figure 4: **Subjective Comparison.** Frame 60 of the "sear_steak" sequence from the Neu3DV dataset and frame 74 of the "softball" sequence from the CMU-Panoptic dataset. We compared our method to Dynamic-GS Luiten et al. (2023) and 4D-GS Wu et al. (2023) over six patches.

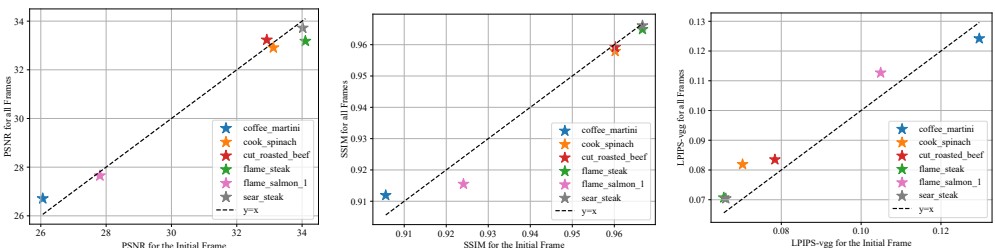

Figure 5: Reconstruction quality correlation between the initial frame and the entire video.

**Subjective Assessment.** We provide subjective comparisons of different sequence patches in Fig. 4. Dynamic-GS struggled with fixed global foreground labels and complex regularization, while 4D-GS had difficulty handling violent motion due to its global deformation field. Our method, however, better preserved details and textures, avoiding issues like excessive smoothing and object disappearance. Subjective results demonstrated that our method excels in preserving and reconstructing details. Comprehensive comparisons are available in the **supplementary video**.

**Significance of Initial 3D Scene.** The poor static scene reconstruction quality in the first frame of the 'coffee_martini' sequence negatively impacted the overall 4D reconstruction. To evaluate the dependence of our 4D reconstruction on the initial 3D scene, we analyzed PSNR, SSIM, and LPIPS-vgg metrics across sequences in the Neu3DV dataset. As shown in Fig. 5, scatter plots with a dashed $y = x$ line indicated similar quality between the initial 3D reconstructions and the overall 4D reconstructions. Our method showed a strong alignment between the reconstruction quality of the video sequence and the initial frame. This indicates that our method's performance is constrained by the initial static scene reconstruction, suggesting that improvements there could enhance overall quality. Additionally, no new Gaussians were added during the sequence, highlighting the efficiency of our method in utilizing the dynamic capabilities of the initial 3D scene.

**Covergance Speed.** Our non-keyframes were optimized in under 2 seconds per frame, and keyframes took approximately 40 seconds each. We anticipate further reductions in processing time as we continue refining the implementation.

## 4.3 ABLATION STUDY

**Points Parameter Comparison.** In Tab. 4, we compared 3D points across various categories to highlight the compactness of our control point method. Each Gaussian is characterized by 13 attributes, including 3D spherical harmonics, position, rotation, scale, and 1D opacity. The number of parameters increases with higher degrees of spherical harmonics.

Table 4: Comparison between Gaussians and control points.

| Points Category | Points Num. | Attr. Dim. | Param. Num. |
|---|---|---|---|
| Scene Gaussians | $> 100k$ | $\geq 13$ | $> 1000k$ |
| Object Gaussians | $\sim 10k$ | $\geq 13$ | $> 100k$ |
| Object Control Points | $0.2k - 2.5k$ | 3 | $0.6k - 7.5k$ |

**Effectiveness of 3D Control Points.** In the GoS-10 configuration, we assessed the rendering quality of three methods: 'No Control' (no motion manipulation), 'Partial Control' (using only projected control points), and 'Full Control' (combining projected and learned control points). PSNR metrics for the first 30 frames of the 'sear_steak' sequence are shown in Fig 6(a). The "No Control" method degraded quickly, "Partial Control" showed moderate degradation, and "Full Control" maintained quality longer. Keyframe updates reduced errors at timesteps 10 and 20. The PSNR gap indicates that integrating projected and learned control points significantly improves reconstruction quality.

**3D Motion Visualization.** We visualized the 3D motion of Gaussians for our method in 6(b). The track precisely described the person turning steaks or getting up to move boxes.

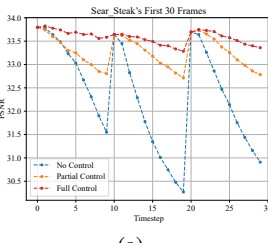 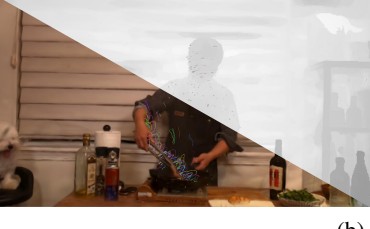 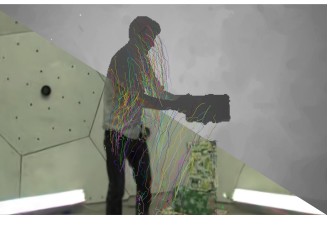

(a)  (b)

Figure 6: (a) An example illustrating reconstruction quality degradation across frames under three different settings of 3D control points. (b) Visualization of Gaussians' 3D motion for the "sear_steak" and the "boxes" sequences.

## 5 CONCLUSION AND DISCUSSION

We introduce a novel discrete 6-DoF motion decoupling model that combines traditional graphics with learnable pipelines. This approach employs partially learnable control points for local 6-DoF motion representation, enabling fast convergence and robust reconstruction for real-world datasets. Additionally, we have developed an innovative workflow for streaming 4D real-world reconstruction using Gaussians and 3D control points. Starting with an initial 3D scene reconstruction, our approach progresses through several independent submodules, allowing each to be optimized individually for future improvements. Our method outperforms existing state-of-the-art 4D Gaussian splatting methods on real-world datasets. The workflow also has limitations: the quality of 4D reconstruction highly depends on the initial static reconstruction, a factor that remains underexplored and presents potential for further research. The current method does not support monocular videos due to its reliance on multi-view initial frames. We plan to address these limitations in future work.

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

# APPENDIX

## .1 NEAR-PARALLEL LIGHT HYPOTHESIS

In the image plane, a small region with center $\mathbf{x}_0$ and the radius $R$ is selected. Then the rays from the projection center connected to each pixel in the region can be regarded as approximate parallel light. We provided a detailed illustration in Fig. 7(a). For an arbitrary $\mathbf{x}_i$ in the $\mathbf{x}_0$'s neighborhood, we should prove that the angle $\theta$ between ray $\mathbf{r}_0$ and ray $\mathbf{r}_i$ is a first-order small quantity.

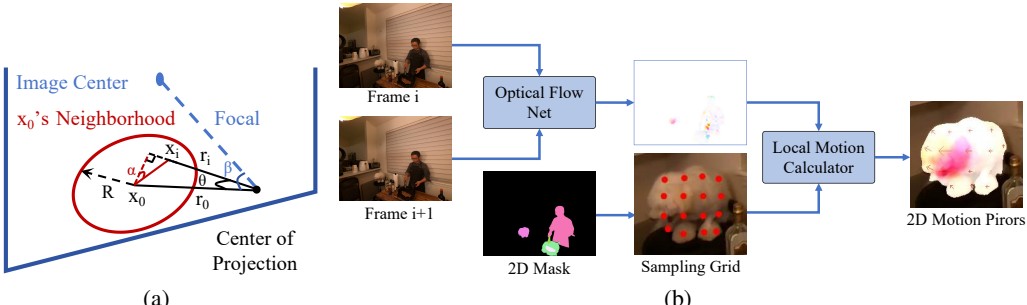

(a)                                          (b)

Figure 7: (a) Illustration for angles, points, rays in $\mathbf{x}_0$'s neighborhood. (b) Workflow for acquiring 2D motion prior.

First, a plumb line is made from $\mathbf{x}_0$ to the ray $\mathbf{r}_i$, which is the shortest path from $x_0$ to the ray and can be denoted as:

$$\|\mathbf{x}_i - \mathbf{x}_0\| \cdot \cos\alpha, \tag{12}$$

where $\|\cdot\|$ stands for the Euclidean distance, and $\alpha$ is the angle between the plumb line and line pointing from $\mathbf{x}_0$ to $\mathbf{x}_i$. Then, using the cosine theorem, the distance from the projection center to $\mathbf{x}_0$ can be represented as:

$$f \cdot 1/\cos\beta, \tag{13}$$

where $f$ is the camera focal length, and $\beta$ is the angle between the ray $\mathbf{r}_0$ and the major optical axis. Next, using the sine theorem, $\theta$ can be characterized as :

$$\theta_i = \mathbf{arcsin}(\frac{\|\mathbf{x}_i - \mathbf{x}_0\|}{f} \cdot \frac{\cos\alpha}{1/\cos\beta}). \tag{14}$$

The second term is always a real number not greater than 1. Since $\|\mathbf{x}_i - \mathbf{x}_0\|$ is always smaller than $R$, $\theta$ remains small if the focal length $f$ is much larger than the neighborhood radius $R$.

Thus far, the proof of Near-parallelism of localized rays is complete. In practice, With a normalized focal length of more than 1000 pixels, it is perfectly acceptable to limit the calculation area to a radius of 50 pixels.

## .2 2D MOTION PRIOR ACQUISITION

We provide an intuitive workflow for the prior acquisition of 3D motion. The inputs consist of two consecutive frames and an object-wise mask.

We first input two consecutive frames into the optical flow network, outputting a whole frame of optical flow. Then, a corresponding sampling grid, based on the objects' masks, is generated for each object in the view. The local motion calculator is the abstract representation of the method we introduced in Sec. 3.3. The final processed 2D motion prior binds with 3D control points.

Note that 3D control points are obtained from all training viewpoints independently. Thus, the 3D control points collect all the training viewpoint control points. Hence, their distribution is much

denser than during single viewpoint control point acquisition, and the range of action of the control points will be smaller. When sampling control points, a larger control point sampling interval should be selected, and for each control point, a smaller range around it should be selected to compute the motion prior.

### .3 SPARSIFICATION FOR 3D CONTROL POINTS

We illustrated the relationship between the Gaussians and the control points in Fig. 8, where the control points are uniformly distributed on the surfaces of objects. Further sparsification of these control points led to notable gains in compactness with only minimal impact on performance.

No Prune:    Human/Dog : 1958/312    Prune 70%:    Human/Dog : 629/94    Prune 90%:    Human/Dog : 212/31
PSNR/SSIM : 33.64/0.9655    PSNR/SSIM : 33.48/0.9645    PSNR/SSIM : 33.42/0.9640

Figure 8: Schematic of Gaussians vs. control points for humans and dogs: We visualized the topology using red and green line segments, with red lines connecting Gaussian points and green lines linking to control points. We also included the number of control points for the first frame of both the human and dog at various pruning rates, alongside the reconstruction quality of the entire sequence.

### .4 3D CONTROL POINTS PRUNE

Further sparsification of 3D control points can be achieved using clustering methods Krishna & Murty (1999); Sinaga & Yang (2020). Note that 3D control points are obtained from all training viewpoints independently. And there exists no prior knowledge of objects' geometry and motion. So we aimed for an even distribution of the sparsified control points. We recommended the k-means++ approach Arthur et al. (2007) due to the large number of clustering centers, which requires a more stable clustering initialization.

Table 5: Comparison between different optical flow methods.

| O.F. Model | Avg. 2D MSE↓ | Rec. PSNR↑ | Rec. SSIM↑ | Rec. LPIPS↓ |
|---|---|---|---|---|
| PWC Sun et al. (2018) | 4.553e-5 | 33.39 | 0.9644 | 0.0737 |
| SpyNet Ranjan & Black (2017) | 1.509e-5 | 33.55 | 0.9649 | 0.0725 |
| DIS Kroeger et al. (2016) | 1.230e-5 | 33.64 | 0.9655 | 0.0716 |

### .5 MORE DETAILED SETTINGS FOR FAIR COMPARISON

**Dynamic-GS Luiten et al. (2023) Setting.**    2D foreground masks and initial 3D points' segmentation labels are required in this approach. To ensure a fair comparison, we merged our objects' 2D masks and used the approach proposed in Sec.3.2 to label the initial points. We shrank the training iterations from 2k to 0.5k per frame when processing the CMU-Panoptic dataset.

**4D-GS Wu et al. (2023) Setting.**    For the "flame_salmon_1" sequence, four times longer than the other sequences, we expanded the training iterations from 17k to 68k to ensure a fair comparison.

### .6 MORE ADVANCED SUBMODULE LEADS TO BETTER RECONSTRUCTION RESULT

We investigated the impact of various optical flow methods Sun et al. (2018); Kroeger et al. (2016); Ranjan & Black (2017) and determined that DIS Kroeger et al. (2016) achieved the most accurate 2D optical flow predictions for Neu3DV dataset. Optical flow accuracy was quantified by the minimal average MSE between the current frame and its warped predecessor. The lowest 2D errors correlated with superior 3D reconstruction quality, suggesting that our pipeline's performance could be further improved by incorporating more advanced optical flow predictors. The positive correlation between optical flow accuracy and reconstruction quality also demonstrates the effectiveness of our 3D motion model.

### .7 MORE SUBJECTIVE RESULTS AT NOVEL VIEWPOINT

We provided more subjective results from different sequences to intuitively evaluate our work. We output the subjective results in the group of two rows, from left to right, top to bottom: GT, Dynamic-GS, 4D-GS, Ours.

Figure 9: More subjective outputs.

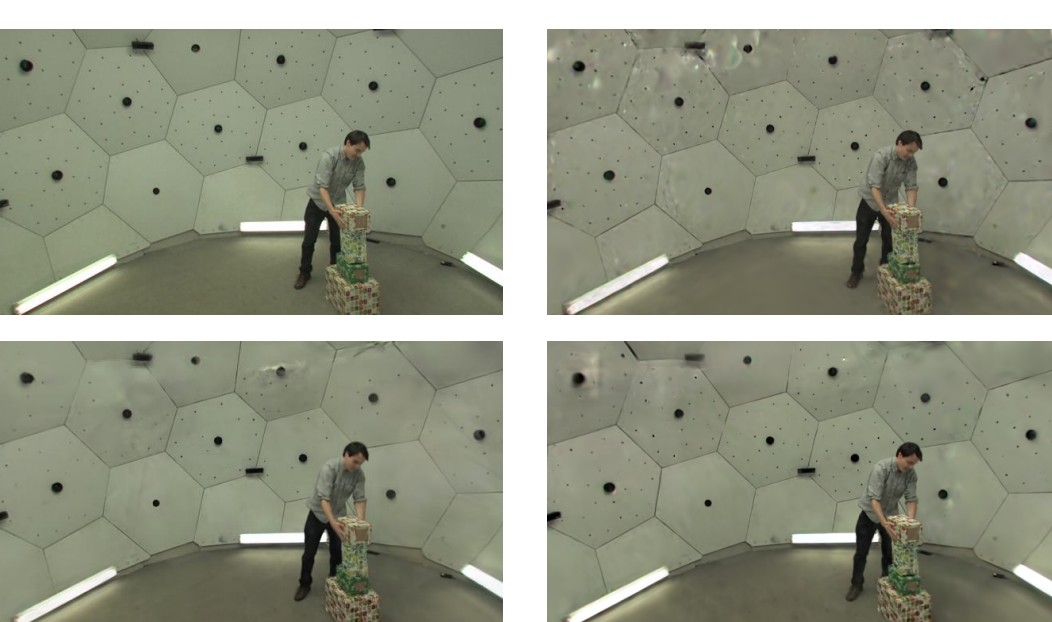

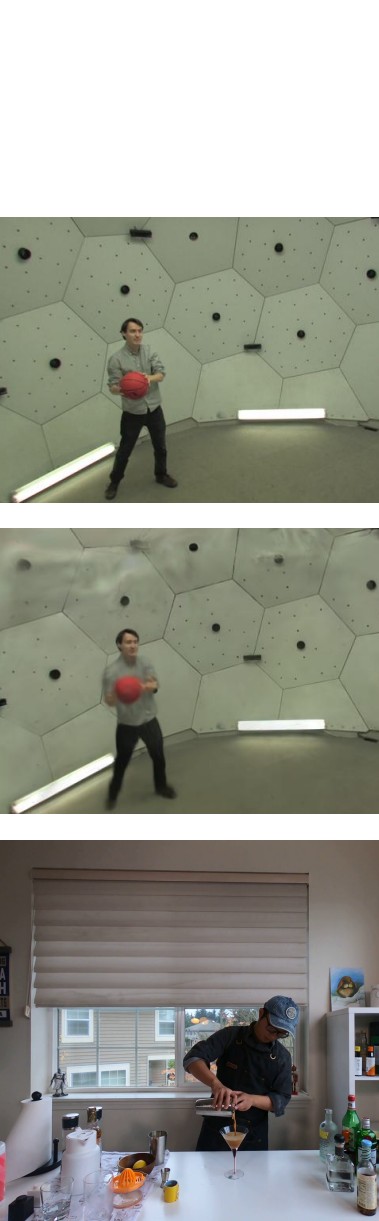
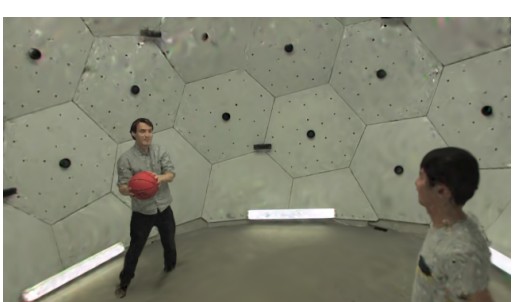
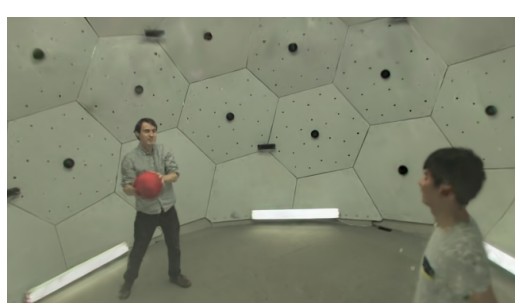
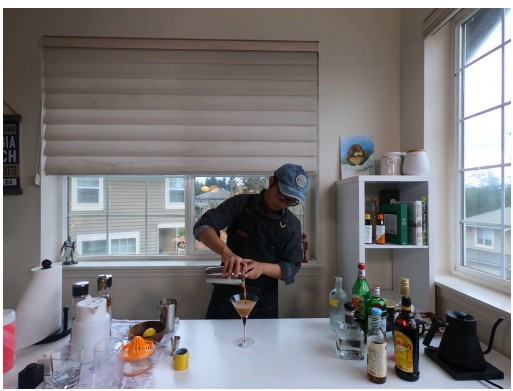
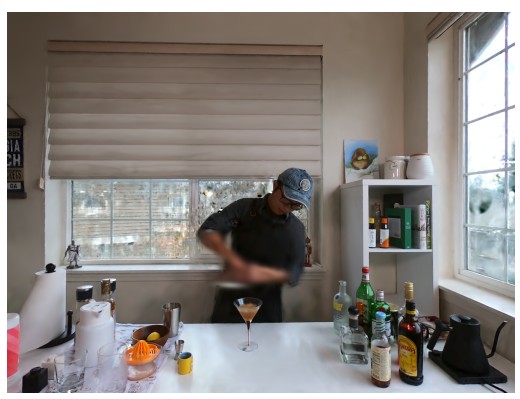
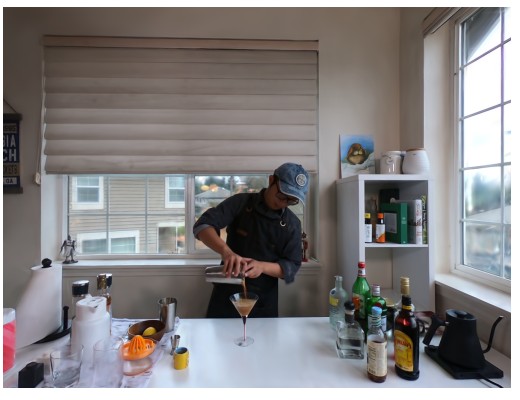
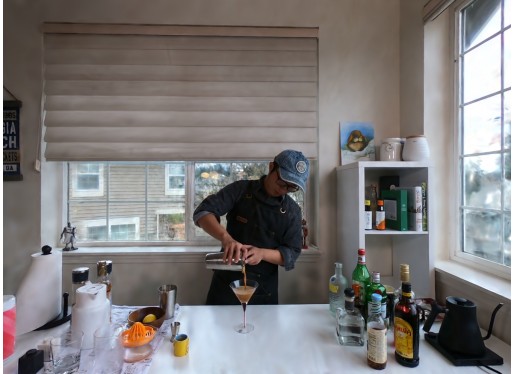

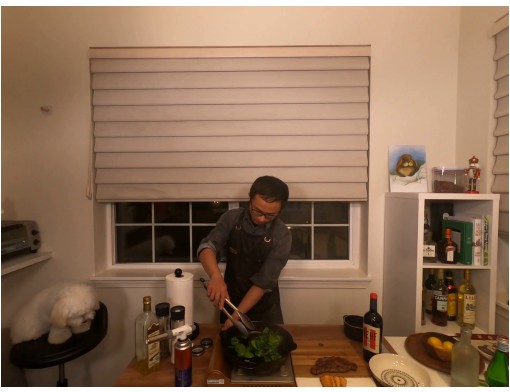
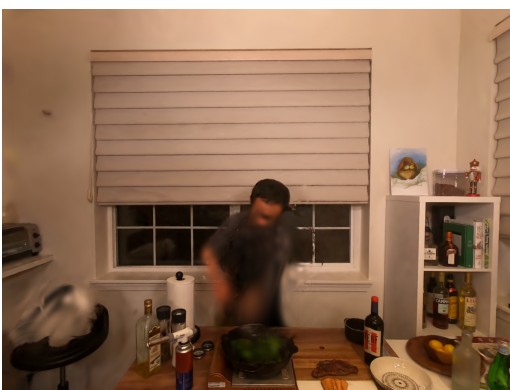

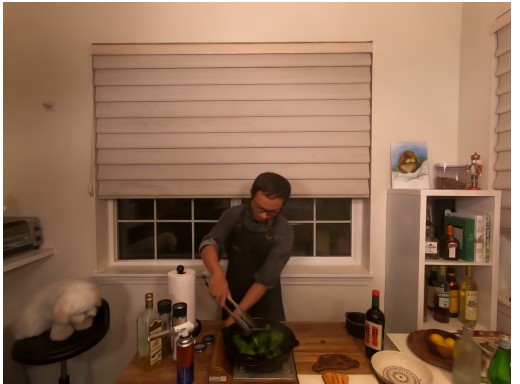
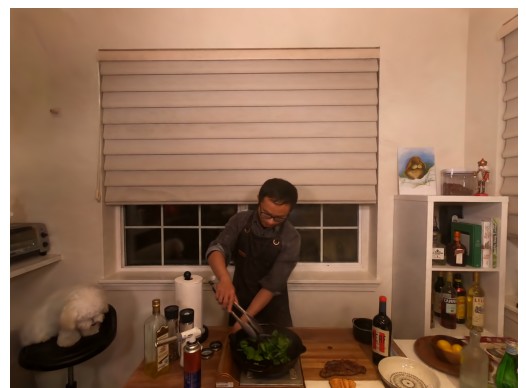

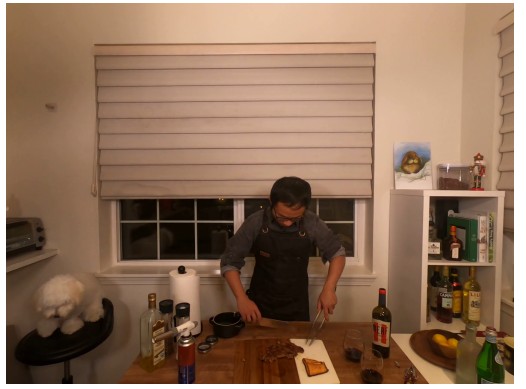
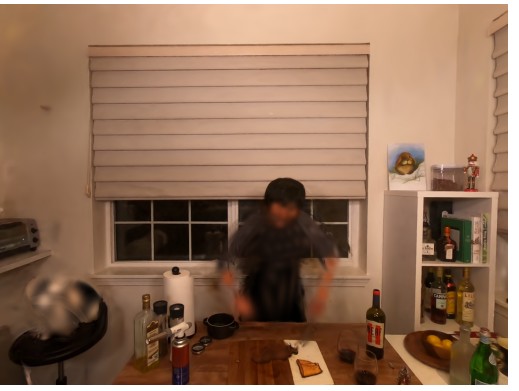

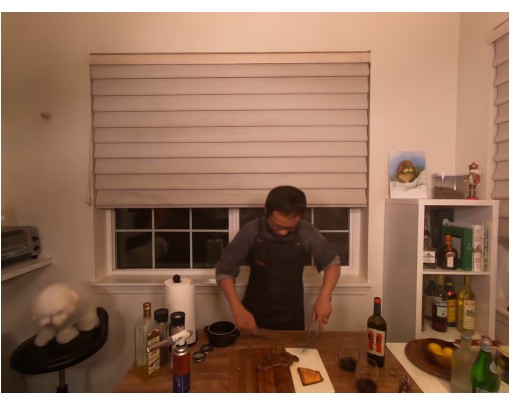
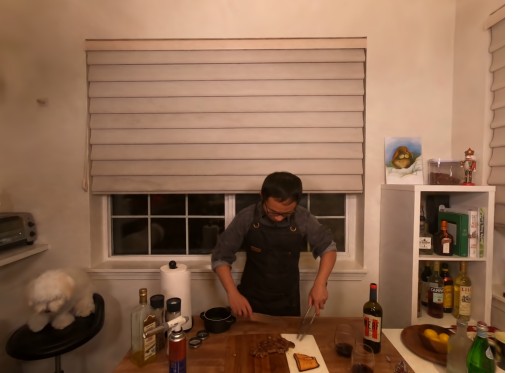

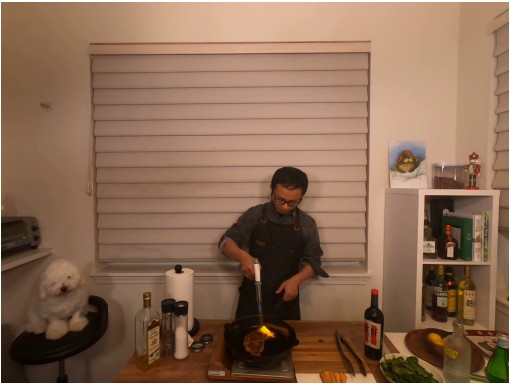 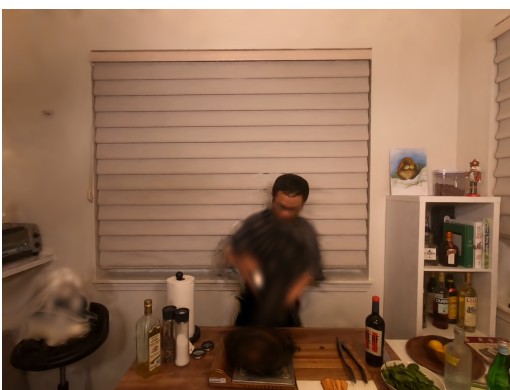

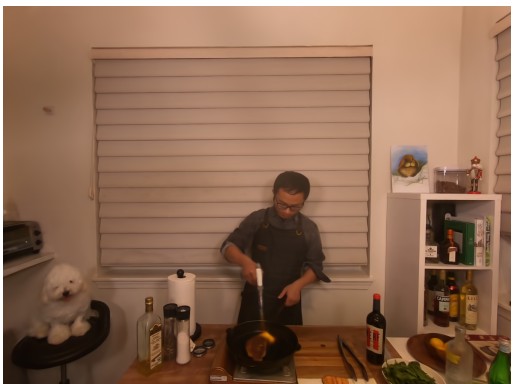 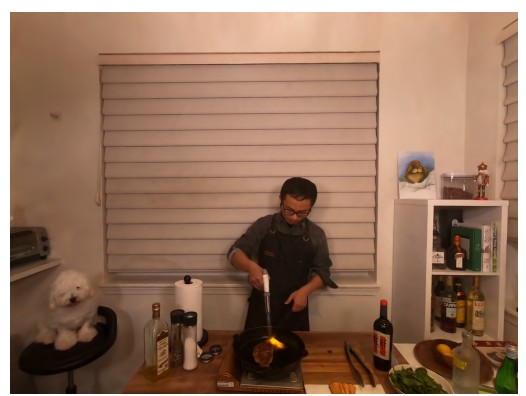

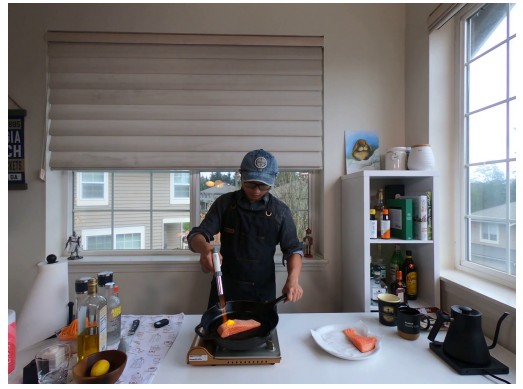 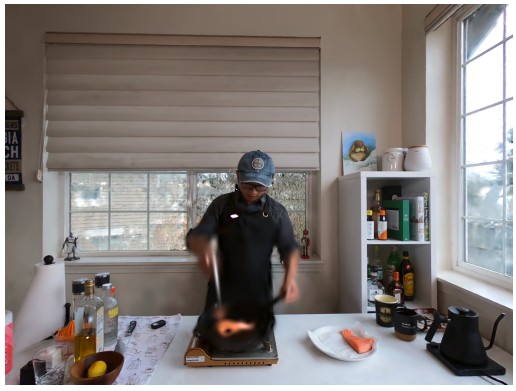

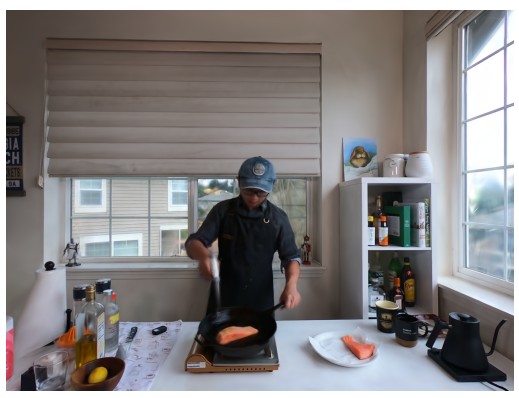 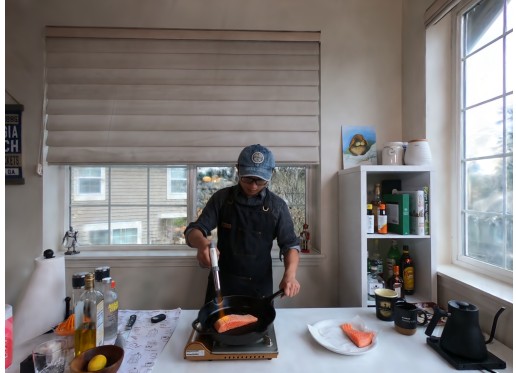

