# OpenReview forum: "S4D: Streaming 4D Real-World Reconstruction with Gaussians and 3D Control Points"
_ICLR.cc/2025/Conference — ICLR 2025 Conference Withdrawn Submission_

### Official Review · Reviewer_Cjxe · 2024-10-26

**Soundness:** 2
**Presentation:** 3
**Contribution:** 3
**Rating:** 5
**Confidence:** 4

**Summary:**

Previous dynamic Gaussian methods typically model the motions by implicit neural networks, whose feature of low-frequency limit the ability to learn complex motions. Besides, the high freedom of  3D Gaussians face difficulties in optimization when modelling the Gaussian's local motion independently. This paper proposes a 3D control point representation to model the dynamic scenes based on 3D Gaussian. The 3D control point's motion is decoupled as the observable part (induced by 2D optical flow) and the invisible part (implicit learned by multi-view training images). Then each Gaussian's motion is modelled as the weighted combination of control point's motion. They achieves the SOTA in Neu3DV and CMU-Panoptic.

**Strengths:**

1. The proposed motion representation is efficient in time and memory.

2. The proposed method can better learn the details and complex motion.

3. The writing is easy to understand.

**Weaknesses:**

1. The motion relys on the estimation of optical flow. I am not sure how robust this method is to errors in optical flow estimation.


2. As authors metioned, the method is strongly relied on the initialized reconstruction and does not densify Gaussians in the subsequent frames. It can not handle the newly emerging objects or occlusion regions.


3. The lack of comparison to the existing methods. SC-GS[1] and SuperPoint-GS[2] share the similar idea with this paper, which use the sparse control points to model the motion. I think the comparison and discussion to these methods are neccesary.


------------------------------------
[1] Huang Y H, Sun Y T, Yang Z, et al. Sc-gs: Sparse-controlled gaussian splatting for editable dynamic scenes[C]//Proceedings of the IEEE/CVF Conference on Computer Vision and Pattern Recognition. 2024: 4220-4230.

[2] Wan D, Lu R, Zeng G. Superpoint Gaussian Splatting for Real-Time High-Fidelity Dynamic Scene Reconstruction[C]//Forty-first International Conference on Machine Learning.

**Questions:**

1. From the paper, it seems that I cannot fully understand how to generate these control points. The author appears to have only introduced the process of generating one control point, but there doesn't seem to be a clear explanation for the generation process of all control points in the entire dynamic area, such as sampling points at what intervals within an image grid. If I have overlooked this part, I apologize.

2. I am confusing about '3D control points are obtained from all training viewpoints independently'. Is this also mean the 3D space contains the 3D control points in different timestamp and each 3D control point only models a transformation in one timestamp?

3. I am not sure that the claim of 'rigid structure of implicit neural networks' in line 40 is adequate. Could authors explain the meaning of it? I understand the smoothness and continuity of neural network, but I am not sure the rigid structure stands for what.

4. In line 137, the authors cite three paper for optical flow estimation. It would be better to cite the paper exactly used in the method and yielded the final experimental results.

5. In Table 1, is the training time reported in hours as stated or is the training time means the total training time for the whole datasets? Since 57.35 hours for Dynamic-GS and 6.88 hours for 4D-GS seems different from their paper.

5. It would be better to anlysis why the proposed method has inferior results in three scenes. Similar like scene 'coffee', authors explain it it because of the poor initialization.

---

### Official Review · Reviewer_YGmv · 2024-11-03

**Soundness:** 2
**Presentation:** 2
**Contribution:** 3
**Rating:** 5
**Confidence:** 4

**Summary:**

The paper introduces an approach to the 4D reconstruction of dynamic scenes using Gaussian splatting and 3D control points. The method combines 4D Gaussian fields with optimized control points to stream reconstructions that are of higher quality than previous frameworks, and robust to changes in lighting and motion.

**Strengths:**

The presented method is novel and has technical merits. Its decoupling of 3D motion into observable and hidden components is an interesting attempt. The generation of control points from optical flow is nicely motivated and discussed.

**Weaknesses:**

Overall, the paper's writing can be further improved, with more concise sentences and better organization. The paper focuses on achieving streamable GS renders in contrast to the setups in the 4DGS line of work, but this point is not clearly stated in the introduction or abstract. The paper currently has an overall presentation of a technical report. It can be further improved if the technical merits, like the motion decoupling using local rays, are highlighted.

Although the system achieves better performance compared to Dynamic-GS, it uses more priors like semantic segmentation masks and optical flow, leading to a longer processing time in practice if accounting for those from the priors. Since processing and convergence time are crucial factors for streaming applications, the paper would be improved if the processing time of these priors were reported and accounted for in the time calculation. Further, the reliance on the priors can be further studied, eg. the behavior of the system if the SAM masks are producing inaccurate segmentations.

**Questions:**

In reference to the Weakness section, clarifications to the following questions would help better understand the system:

1. How much time do the segmentation and optical flow priors add to the system? Are the convergence times in the paper reported excluding the prior times?
2. Why are there three sources of optical flow priors? How are they chosen and what is the rationale for picking them?
3. The baseline comparisons included are Dynamic-GS and 4DGS, both of which do not rely on segmentation or motion priors. How does the proposed approach compare to other dynamic Gaussian Splatting papers utilizing a similar set of priors?
4. Sec. 3.5 presents an interesting approach to mitigate accumulated errors. However, with the current presentation, the use of Group of Scenes (GoS) is not sufficiently motivated. Why is it necessary to adopt this similar formulation to Group of Pictures? And how does residual coding reduce accumulated error? These novelties should deserve more elaboration.
5. The concept of control points has been explored in other literature like ScaffoldGS - although formulated differently. How does the proposed control point formulation compare to these previous formulations, either conceptually or performance-wise?
6. How would the system behave given inaccurate/noisy segmentation priors or flow priors?

---

### Official Review · Reviewer_UUXw · 2024-11-03

**Soundness:** 2
**Presentation:** 2
**Contribution:** 2
**Rating:** 5
**Confidence:** 4

**Summary:**

This paper study how to photometrically reconstruct the 4D scene from multi-view calibrated videos.  The pipeline first use 3D segmentation to segment object patches from mutliview, then use the local optical flow patches to construct the SE(3) control points, finally the gaussian is deformed by skinning inside each object patch. Notably, the system is an online method that depends on the previous frame reconstruction to compute the new incoming frame. Comparison with other GS based methods is conducted on dynerf and panoptic dataset.

**Strengths:**

- On a high-level, using skinning and motion basis is a correct way in a long term to model real world 4D motion.
- Including object patch wise modeling is a correct way to simplify 4D reconstruction.
- The system is online, which enables longer reconstruction and streaming, however, it also has significant drawback (see weakness)

**Weaknesses:**

- Because the system processes the frame one by one, later information or observations cannot be fused into previous reconstruciton, which may lead to two possible situations, either the system depends on super good initialization from a multiview camera setup, or the earlier reconstruction will have larger error, and such error will accumulate as the online reconstruction goes to later frames. There is no evidence how the proposed method deals with such accumulated error when streaming to very long video, and how robust it is if one frame in the middle fail.
- Occlusion: the key challenge of 4D reconstruction is not what is observed (visible optical flow), instead, is what is temporally unobserved, it’s not clear that how the proposed control point construction method will behave when the object is largely occluded, this is not verified in the experiment because the dataset used in the experiments are very easy and widely studied ones.
- The experiments comparison is a little limited to only GS based methods, and there is no clear argument why the visual results are improved because of any novel design proposed in the paper.
- The writing and presentation is a little wired, in the method part, there is a very long overview where I mistake as the main method paragraphs at the beginning, also the English writing sounds a little over-polished by LLM.
- It’s also not clear that how robust the Multiview VOS is and how it affects the performance of the system, because if not monitored by human, the segmentation mask may have small noise or miss segmentation in real world videos. Also, it’s unclear how to make consistent segmentation level/granularity
- The related works is does not fully cover the most recent advances of 4D reconstruction, many recent papers are not included.

**Questions:**

Please see the weakness for details, the main question is to justify the robustness of the proposed online paradigm in more challenging (maybe fewer camera, other harder videos) cases, how it behaves if a middle frame fails? if the segmentation is not accurate? and expand to include a few more baselines.

---

### Official Review · Reviewer_5h3T · 2024-11-11

**Soundness:** 2
**Presentation:** 1
**Contribution:** 2
**Rating:** 3
**Confidence:** 5

**Summary:**

This paper presents a pipeline for 4D novel view rendering.

The pipeline starts with a segmentation of 3D Gaussians into foreground objects and background. 3D control points are created as centers of regions modeled with locally parallel rays.
Then, these 3D control points are propagated over time through the motion manipulation modules. Finally, keyframes are defined and used for the optimization of Gaussian parameters and control points.

The results are impressive. I watched the sear_steak and cool_spinach, and the local details (white dog, etc) are really remarkable.

**Strengths:**

+ Results are impressive both in terms of quantitative comparison and subjective experience.

+ The idea of control points is really promising.

+ The diagram in Fig.1 is clear.

**Weaknesses:**

I watched the video first, and then I glimpsed through the experimental results. I was very impressed and eager to read the methodology of the paper. While the main diagram in Fig. 1 was clear, the description of the individual components was very confusing.

W1. The main innovation of the method (and based on the ablation having a large effect) is the control points. However, at no point in the paper, and not even in 3.3, is there any definition of how a control point is defined. There is a description of what is done with the control points but not what is a control point and what is not a control point.

W2. Dynamic reconstruction from multiple static views needs constraints for regularization: either a reduction to a canonical space or a prior for trajectories in terms of a neural prior or a trajectory basis. In this paper, it is not clear whether such a regularization comes from the segmentation and the object level manipulation or from the residual compensation.

W3: Line 211 and Appendix 1. It is a well-known fact that when the FOV is small the projection can be approximated with a scaled orthographic projection. The authors' calculation has to be set in the right textbook context. The alignment of the optical axis with the central ray of the neighborhood is a well-known fact in the approximation of small object projection with scaled orthograph (see Hartley and Zisserman, for example)

W4: The paper does not have a flow on the order of computation of things. If one tries to go backward from (10) , we have scenes $\mathbb G_t$ that are propagated with a function $\mathcal F_{\mathbb C_t}$ that is nowhere defined. We guess that this is somehow related to (t,q) in (6). Then, there is no formula for computing (t_i,q_i) for control points except an angular velocity formula in (5) or linear velocity in (3).

W5: The computation of linear and angular velocity in a frame centered at the target ray where scaled orthography is assumed can be computed with by transferring the classic optical flow formula for scaled orthography to a different frame. However, it is not clear how only the []:2:2 submatrix of rotation reflects the rotation of the optical axis to the target ray. If for example the target ray is along the x-axis, then the rotation is around the y-axis [ cos & 0 & sin \\ 0 & 1 & 0 \\ -sin &0 & cos] and the submatrix  of the form [ cos ; 0 \\ 0 ; 1] cannot capture the full rotation. You might be correct for the scaled orthography case but please explain.

W6: L_1 and L_{D-SSIM} should be defined. You either say "We use the the loss of the original Gaussian splitting paper..." or you define your symbols.

W7: Several dynamic Gaussian CVPR papers have not been cited:
CVPR24: Control4D: Efficient 4D Portrait Editing with Text
CVPR24: Neural Parametric Gaussians for Monocular Non-Rigid Object Reconstruction
CVPR24: CoGS: Controllable Gaussian Splatting
CVPR24: Spacetime Gaussian Feature Splatting for Real-Time Dynamic View Synthesis

W8: There is a "double" citation in the paper
Youtian Lin, Zuozhuo Dai, Siyu Zhu, and Yao Yao. Gaussian-flow: 4d reconstruction with dynamic 3d gaussian particle. arXiv preprint arXiv:2312.03431, 2023b.

Youtian Lin, Zuozhuo Dai, Siyu Zhu, and Yao Yao. Gaussian-flow: 4d reconstruction with dynamic 3d gaussian particle. In Proceedings of the IEEE/CVF Conference on Computer Vision and Pattern Recognition, pp. 21136–21145, 2024.

W9: 3.5 is highly confusing, in particular, because of the introduction of calligraphic and mathbb fonts. If put in a classic context this is nothing else than error state propagation equation in the Kalman filter with the resisual being the control input.

W10: L. 53: ". Recognizing that optical flow is the 2D projection of scene flow," This is wrong. Optical flow (technically the motion field not optical flow) is not the projection of scene flow. $$({\\dot x},{\\dot y}) \\neq \\frac{1}{Z} (({\\dot X},{\\dot Y},{\\dot Z})$$. Please see a classic textbook like Berthold Horn.

W11: L. 53: Optical flow is a well studied field in classic articles from the 1980's and it is textbook material. It does not start with Ranjan and Black in 2017.

**Questions:**

Q1: In line 53, you write, "Our approach differs by integrating graphics and learnable pipelines." There is nothing learnable in this paper, just the usual optimization for one scene.  I guess by graphics you mean that you use the rendering equation.

Q2: Rotation parametrization is confusing. In L. 286 you say that you need Euler angles. The angular velocity is just a velocity and does not have anything to do with the Euler angles unless you use the derivatives of Euler angles (see end of https://phys.libretexts.org/Bookshelves/Classical_Mechanics/Classical_Mechanics_(Tatum)/04%3A_Rigid_Body_Rotation/4.02%3A_Angular_Velocity_and_Eulerian_Angles).

Q3: L199 is not an algorithm (there is no for loop or end condition). Wouldn't it be simpler to describe it in a sentence?

Q4: L174: "This selective control method allows for accurate handling of topology changes". Is it meant here the relation between object parts or just local topology changes (contact etc) ?

Q5: L207: What do you mean by "advanced" in a coordinate system?

Q6: 3.4 and eqs (6-7): It is not clear whether rigid motion interpolation can be achieved with linear interpolation of quaternions. Computer graphics has shown that linear blend skinning brings a lot of artifacts. A justification is necessary.

Q7: L. 867 Why was no pre-trained RAFT used for optical flow which is the state of the art?

Q8: Please rely to all weaknesses.

---

### Author Response · Authors · 2024-11-15
**Thanks for all the responsible and insightful feedback!**

Thx

---

### Note · Authors · 2024-11-15

I have read and agree with the venue's withdrawal policy on behalf of myself and my co-authors.